# ICOS costimulation is indispensable for the differentiation of T follicular regulatory cells

Vincent Panneton[1,2], Barbara C Mindt[3,4], Yasser Bouklouch[1], Antoine Bouchard[1,5], Saba Mohammaei[1,6], Jinsam Chang[1,5], Nikoletta Diamantopoulos[1,3], Mariko Witalis[1,5], Joanna Li[1,3], Albert Stancescu[1], John E Bradley[7], Troy D Randall[7], Jörg H Fritz[3,4], Woong-Kyung Suh[1,2,3,5,6]

ICOS is a T-cell costimulatory receptor critical for Tfh cell generation and function. However, the role of ICOS in Tfr cell differentiation remains unclear. Using Foxp3-Cre–mediated ICOS knockout (ICOS FC) mice, we show that ICOS deficiency in Treg-lineage cells drastically reduces the number of Tfr cells during GC reactions but has a minimal impact on conventional Treg cells. Single-cell transcriptome analysis of Foxp3$^+$ cells at an early stage of the GC reaction suggests that ICOS normally inhibits *Klf2* expression to promote follicular features including *Bcl6* upregulation. Furthermore, ICOS costimulation promotes nuclear localization of NFAT2, a known driver of CXCR5 expression. Notably, ICOS FC mice had an unaltered overall GC B-cell output but showed signs of expanded autoreactive B cells along with elevated autoantibody titers. Thus, our study demonstrates that ICOS costimulation is critical for Tfr cell differentiation and highlights the importance of Tfr cells in maintaining humoral immune tolerance during GC reactions.

## Introduction

High-affinity class-switched antibodies are essential for immune responses against pathogens. These antibodies arise from germinal centers (GCs) where T follicular helper (Tfh) cells facilitate the transition of GC B cells into antibody-secreting plasma cells (PCs) (Crotty, 2011; Victora & Nussenzweig, 2012). Tfh cells are defined by the combined expression of their master transcription factor Bcl6 along with CXCR5, PD-1, and ICOS (Fazilleau et al, 2009; Johnston et al, 2009; Nurieva et al, 2009; Yu et al, 2009). Tfh cells mediate their helper functions through costimulation by CD40L and ICOS along with the production of the cytokines IL-4 and IL-21 (Bryant et al, 2007; Reinhardt et al, 2009; Crotty, 2014). Because dysregulation of

Tfh cells and GC reactions can lead to humoral autoimmunity, they must be tightly controlled (Linterman et al, 2009).

T follicular regulatory (Tfr) cells are a subset of CD4$^+$ Foxp3$^+$ regulatory T (Treg) cells found in and around germinal centers (Chung et al, 2011; Linterman et al, 2011; Wollenberg et al, 2011; Sayin et al, 2018). Like Tfh cells, they express the chemokine receptor CXCR5, which is required for their migration towards B-cell follicles (Chung et al, 2011; Linterman et al, 2011; Wollenberg et al, 2011). The transcription factor NFAT2 was recently shown to be required for CXCR5 up-regulation by Tfrs, possibly to overcome BLIMP-1–mediated CXCR5 down-regulation (Shaffer et al, 2002; Oestreich et al, 2012; Vaeth et al, 2014). There are no known lineage-defining factors specific to Tfr cells, although they require the concomitant expression of Foxp3 and Bcl6 (Wu et al, 2016; Hou et al, 2019). A significant proportion of Tfr cells originate from thymic Tregs and possess a T-cell receptor (TCR) repertoire skewed towards self-antigens (Chung et al, 2011; Linterman et al, 2011; Wollenberg et al, 2011; Maceiras et al, 2017). Under specific conditions, induced Tregs have displayed the capacity to differentiate into Tfr cells that can be specific for the immunizing antigen (Aloulou et al, 2016). Strong IL-2 signaling was shown to inhibit Tfr differentiation, which is more akin to Tfh cells rather than Tregs (Botta et al, 2017). Although the in vivo role of Tfr cells has been controversial, they display suppressive abilities on T-cell proliferation, antibody secretion, and cytokine production in vitro (Sage et al, 2013, 2014). Several Tfr depletion models have been studied to understand their functions in vivo. Initially, adoptive transfer or mixed BM chimera experiments showed that Tfr reduction had varying effects on GC responses, possibly because of unintended side effects such as impaired Treg function (Chung et al, 2011; Linterman et al, 2011; Wollenberg et al, 2011). More recently, Foxp3-specific deletion of Bcl6-expressing (Bcl6 FC) or CXCR5-expressing (Tfr-deleter) cells allowed for a more precise assessment of in vivo roles of Tfr cells (Wu et al, 2016; Laidlaw et al, 2017; Fu et al, 2018; Xie & Dent, 2018; Clement et al, 2019; Lu et al, 2021). Results from these studies collectively suggest two

[1]Institut de Recherches Cliniques de Montréal, Quebec, Canada   [2]Department of Microbiology, Infectiology and Immunology, University of Montreal, Quebec, Canada   [3]Department of Microbiology and Immunology, McGill University, Quebec, Canada   [4]McGill University Research Centre on Complex Traits, McGill University, Quebec, Canada   [5]Molecular Biology Program, University of Montreal, Quebec, Canada   [6]Division of Experimental Medicine, McGill University, Quebec, Canada   [7]Department of Medicine, Division of Clinical Immunology and Rheumatology, University of Alabama at Birmingham, Birmingham, AL, USA

Correspondence: woong-kyung.suh@ircm.qc.ca

roles of Tfr cells: suppression of autoantibody production and modulation of antibody responses more suggestive of "helper" functions (Sage & Sharpe, 2020).

The inducible costimulator (ICOS) is a member of the CD28 superfamily and is known to be expressed by activated T cells (Hutloff et al, 1999). ICOS was previously shown to be essential for the formation of Tfh cells and maintenance of Bcl6 expression (Bossaller et al, 2006; Gigoux et al, 2009; Leavenworth et al, 2015). In both mice and humans, ICOS null mutations cause severe defects in GC reactions and antibody production because of the lack of Tfh cells (Dong et al, 2001; McAdam et al, 2001; Tafuri et al, 2001; Grimbacher et al, 2003). Some ICOS-deficient patients develop autoimmune symptoms such as rheumatoid arthritis and auto-immune neutropenia, suggesting a potential role of ICOS in Treg/Tfr compartments (Warnatz et al, 2006; Takahashi et al, 2009). Indeed, ICOS deficiency in mice led to reduced Tfr cell numbers, although the underlying mechanisms have not been carefully analyzed (Sage et al, 2013; Zhang et al, 2018).

In this study, we used $Icos^{fl/fl}$ $Foxp3$-$Cre$ (ICOS FC) mice to evaluate the role of ICOS signaling in Treg and Tfr cells during GC reactions. Foxp3-specific loss of ICOS led to a significant decrease in Tfr populations after protein immunization or viral infection without affecting Treg cell numbers. Examination of antibody responses revealed significantly lowered IgG2b titers at steady state or after immune challenge along with increased anti-nuclear autoantibodies in ICOS FC mice. Single-cell transcriptomics and biochemical analyses suggest that ICOS may enhance the Treg-to-Tfr transition through KLF2 and NFAT2 regulation. Overall, our findings indicate that the major role of ICOS in regulatory T-cell compartments during GC reactions is to control Tfr differentiation, and highlight the importance of Tfr cells in preventing autoantibody generation.

# Results

## ICOS is required for Tfr cell generation during GC reactions against protein antigens

To assess the role of ICOS in Treg-lineage cells, we used a Foxp3-Cre system, which allows for the specific abrogation of ICOS expression in all Treg and Tfr cells. Throughout this study, we used $Icos^{+/+}$ $Foxp3$-$Cre^+$ controls (ICOS WT) for $Icos^{fl/fl}$ $Foxp3$-$Cre^+$ mice (ICOS FC). First, we analyzed splenocytes 12 d after immunization with NP-OVA/alum by flow cytometry and subdivided the CD4$^+$ Foxp3$^+$ regulatory T-cell compartment into Treg (CD4$^+$ Foxp3$^+$ CXCR5$^-$ PD-1$^-$), PD-1$^-$ Tfr (CD4$^+$ Foxp3$^+$ CXCR5$^+$ PD-1$^-$), and PD-1$^+$ Tfr (CD4$^+$ Foxp3$^+$ CXCR5$^+$ PD-1$^+$) subsets (Fig 1A). ICOS was expressed in both Tfh (CD4$^+$ Foxp3$^-$ CXCR5$^+$ PD-1$^+$) and Treg/Tfr cells, with PD-1$^+$ Tfr cells showing the highest surface levels (Fig S1A).

We confirmed that ICOS deletion was limited to Foxp3$^+$ cells and did not occur in Tfh cells (Fig S1B). We observed no change in Treg cell numbers in ICOS FC mice (Fig 1A). However, we found ~twofold and ~fourfold decreases in PD-1$^-$ and PD-1$^+$ Tfr cell proportions and numbers, respectively. We also observed a similar decrease in PD-1$^+$ Tfr proportions in unimmunized ICOS FC mice (Fig S2). Furthermore,

our histological analysis revealed that Foxp3-specific ICOS abrogation in NP-OVA/alum-immunized mice leads to a ~fourfold reduction in Foxp3$^+$ cells within the GC (Fig S3). Next, we examined Tfh, GC B, and plasma cells because a lack of Tfr cells has been shown to increase these populations in some experimental settings (Chung et al, 2011; Linterman et al, 2011; Wollenberg et al, 2011). We did not observe significant differences in Tfh cell proportion or absolute numbers (Fig 1A). We also detected no quantitative differences in GC B-cell and plasma cell populations (Fig 1B and C). Consistently, NP-specific IgG1 and IgG2b antibodies in serum did not show changes in total (NP30) or high-affinity (NP7) titers (Fig 1D and E). Taken together, these results indicate that loss of ICOS in Foxp3$^+$ cells leads to a specific reduction in Tfr populations.

## ICOS is required for Tfr generation during anti-viral responses

To evaluate the role of ICOS in regulatory T-cell compartments during an anti-viral immune response, we infected mice with influenza A virus (IAV). IAV infection experiments using Foxp3-specific Bcl6 knockout mice have shown altered GC B-cell responses in the absence of Tfr cells (Lu et al, 2021). We analyzed splenocytes by flow cytometry 30 d after infection because it has been shown that Tfr generation is delayed because of high levels of IL-2 present in the early stages of viral infection (Botta et al, 2017; Lu et al, 2021). We observed no change in Treg cell numbers and decreases in Tfr populations in spleens of ICOS FC mice reminiscent of results from protein immunization experiments (Fig 2A). As before, we did not detect significant differences in Tfh populations (Fig 2B). Next, we examined the expansion of GC B-cell populations that recognize the IAV nucleoprotein using recombinant tetramers (Flu tetramer) (Allie et al, 2019). Interestingly, we observed a trend of increased total GC B cells with a significantly decreased proportion of IAV-specific GC B cells in ICOS FC mice (Fig 2C). These results strongly suggested an increase in non–IAV-specific GC B cells, some of which could be autoreactive in nature. On the contrary, we did not observe significant differences in splenic plasma cells (Fig 2D). However, we found that the O.D. values for IAV-specific IgG2b antibodies were significantly reduced in ICOS FC mice, suggesting lower antibody titers and/or affinity (Fig 2E). Consistent with the role of anti-viral antibodies in the overall control of influenza virus (Lam & Baumgarth, 2019), we observed that ICOS FC mice had more severe weight loss 9–10 d post-infection (Fig S4A). This reduced anti-viral IgG2b titer was well correlated with a reduction in total IgG2b titers in infected ICOS FC mice (Fig S4B). Furthermore, we noticed that uninfected ICOS FC mice had decreased basal levels of IgG2b and IgG1 (Fig S4C), contrasting normal anti-NP IgG1 and IgG2b titers shown in NP-OVA/alum immunization experiments (Fig 1E). These data suggest that ICOS-expressing Treg or Tfr cells may play differential roles in humoral immunity depending upon the antibody isotype and the immunological settings. Nonetheless, largely congruent with data from protein immunization experiments, this infection model confirmed the critical role of ICOS in efficient Tfr generation. Furthermore, increases in non–viral-specific GC B cells in ICOS FC mice confirm the regulatory role of Tfr cells in shaping GC responses, consistent with previous observations (Lu et al, 2021).

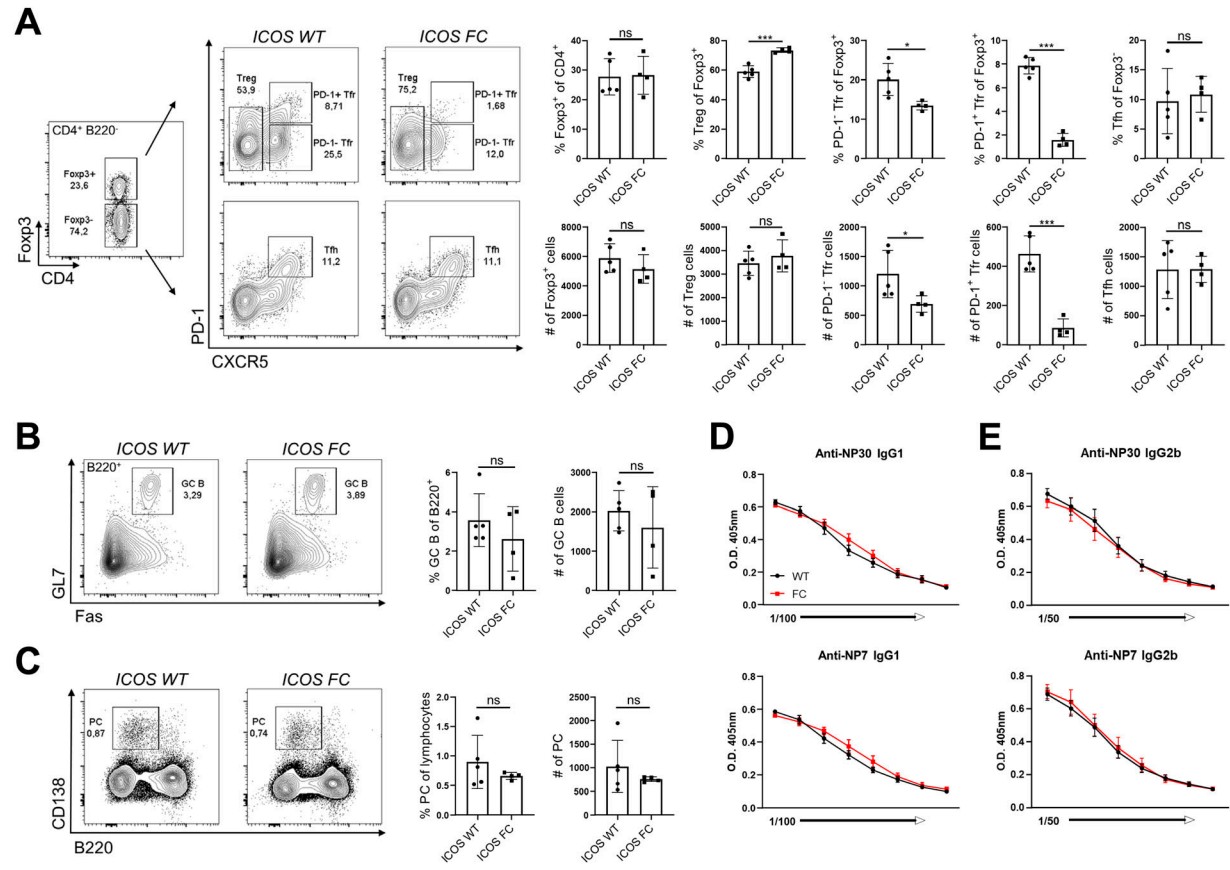

**Figure 1.   Foxp3-specific ICOS ablation decreases Tfr populations in a protein immunization model.**
Splenocytes from ICOS WT (n = 5) and ICOS FC (n = 4) mice were harvested 12 days post-immunization (dpi) with NP-OVA/alum and analyzed by flow cytometry. **(A)** Treg, Tfr, and Tfh cell percentages and numbers were evaluated using a combination of Foxp3, PD-1, and CXCR5 staining. **(B, C)** B220+ Fas+ GL7+ GC B-cell populations and (C) B220− CD138+ plasma cell populations were analyzed from the same splenocyte pool 12 dpi. **(D, E)** Total (NP30) and high-affinity (NP7) anti-NP IgG1 titers (D) and IgG2b titers (E) were measured by ELISA using serum from ICOS WT (black, n = 5) or ICOS FC (red, n = 5) mice obtained 28 dpi. All the serum samples underwent twofold serial dilutions starting from 1:100 (IgG1) or 1:50 (IgG2b). Data are shown as the mean ± SEM, *$P < 0.05$, **$P < 0.01$, and ***$P < 0.001$. All data are representative of at least three independent experiments.

## Treg-specific ICOS deficiency leads to anti-nuclear antibody production

Because there is ongoing cell death and release of autoantigens within the GC, autoreactive GC B-cell clones can expand and differentiate into PCs with help from Tfh cells if not restrained (Linterman et al, 2009). Studies have found that Tfr-deficient mice fail to suppress self-reactive antibody production (Fu et al, 2018; Lu et al, 2021). Given that Foxp3-specific ICOS ablation results in reduced numbers of Tfr cells, we investigated whether ICOS FC mice displayed signs of autoimmunity. We did not observe immune infiltration in the kidneys, lungs, spleen, pancreas, and salivary glands of 5-mo-old ICOS FC mice (Fig 3A). Next, we used HEp-2 slides to look for the presence of anti-nuclear antibodies (ANAs), which are a hallmark of autoimmunity (Fig 3B) (Castro & Gourley, 2010). We did not detect significant spontaneous increases in ANAs in serum samples of 6-mo-old ICOS FC mice. However, we found that both single NP-OVA/alum immunization and secondary challenge with the same antigen resulted in significantly higher ANAs in ICOS FC mice. In addition, ICOS FC mice infected with IAV presented similar increases in ANA levels. These results

suggest that immunization or infection augments adventitious generation of autoantibodies, which is normally suppressed by Tfr cells.

## ICOS-deficient Treg cells show impaired transition to Tfr cells

To better understand the role of ICOS in Tfr cell differentiation, we performed single-cell transcriptome analysis of CD4+ Foxp3+ splenocytes sorted from ICOS WT and ICOS FC mice immunized with NP-OVA/alum. To collect cells in a dynamic Treg-to-Tfr transition stage, we prepared samples 6 d post-immunization, a timepoint where Tfr cells begin to appear (Fig S5A) (Sage et al, 2013; Botta et al, 2017). After sorting, we added back-sorted conventional CD4+ Foxp3− T cells (~10% of total) to the sorted CD4+ Foxp3+ T-cell pool to provide a reference point for Foxp3-null cells (Fig S6A, cluster 4). Our flow cytometry data indicated that the proportion of Tfh cells in the Foxp3− reference population is about 2% at day 6 post-immunization (Fig S5B). Therefore, only 0.2% of our single-cell data points would represent Tfh cells, which should not affect the overall population features of Treg and Tfr cells. We used Tfr-defining genes (*Cxcr5, Pdcd1, Foxp3, Bcl6*) to calculate a "Tfr identity

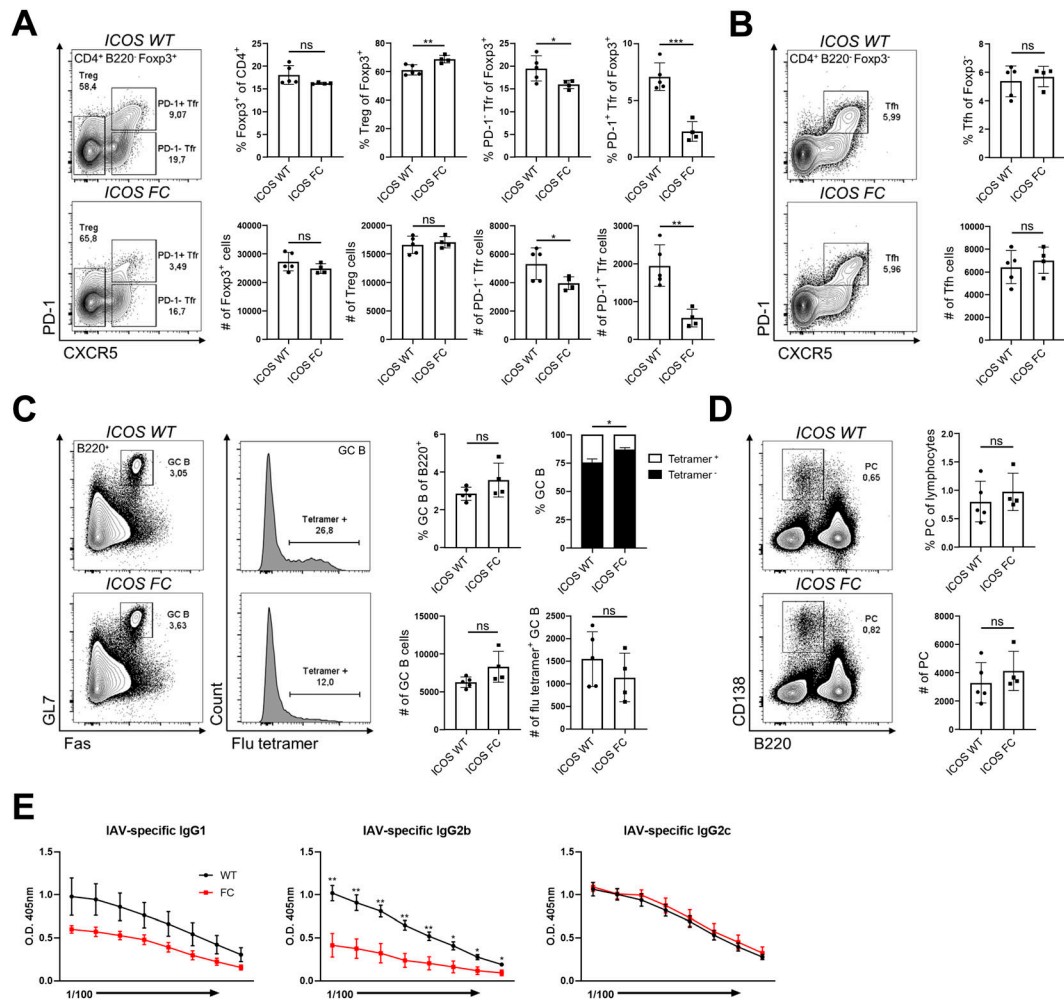

**Figure 2. ICOS FC mice display reduced Tfr populations along with increased extraneous GC B cells during anti-viral responses.**
Splenocytes from ICOS WT (n = 5) or ICOS FC (n = 4) were harvested 30 days post-infection (dpi) with influenza A virus and analyzed by flow cytometry. **(A)** Foxp3$^+$ Treg and Tfr cells were classified using CXCR5 and PD-1 staining. **(B)** Foxp3$^-$ CXCR5$^+$ PD-1$^+$ Tfh cells were analyzed from the same splenocyte pool. **(C)** B220$^+$ Fas$^+$ GL7$^+$ GC B cells were harvested from spleens 30 dpi and stained with influenza nucleoprotein–specific tetramers (flu tetramers). **(D)** B220$^-$ CD138$^+$ plasma cells were analyzed using the same splenocyte pool. **(E)** Influenza A virus–specific IgG1, IgG2b, and IgG2c titers were measured by ELISA using serum samples from ICOS WT (black) and ICOS FC (red) mice obtained 30 dpi. All the serum samples underwent twofold serial dilutions starting from 1:100. Data are shown as the mean ± SEM, *$P$ < 0.05, **$P$ < 0.01, and ***$P$ < 0.001. All data are representative of two independent experiments.

score" and selected three clusters that are predicted to contain Tfr precursors and mature Tfr cells (Fig S6B, clusters 3, 5, and 8, black arrows). When compared among each other, these cells formed three distinct clusters with a spectrum of Tfr identity score (Fig 4A, clusters 1, 2, and 3; equivalent to clusters 3, 5, and 8 in Fig S6B, respectively). Pseudotime trajectory analysis revealed a progressive differentiation from cluster 1 towards cluster 3 (Fig 4B). Interestingly, we observed that ICOS FC mice presented a threefold increase in cells in cluster 2 and a threefold decrease in cluster 3 (Fig 4A). This suggests that Tregs could be halted in their transition to Tfr cells because of the loss of ICOS expression. To substantiate this idea, we compared gene expression profiles of the three clusters (Fig 4C, left). Cluster 3 had the highest levels of key Tfr signature genes (*Cxcr5*, *Pdcd1*, and *Bcl6*), but reduced expression of typical Treg signature genes such as *Foxp3*, *Il2ra* (CD25), *Ctla4*, and *Tnfrsf18* (GITR) when compared to clusters 1 and 2. However, when

compared to CD4$^+$ Foxp3$^-$ conventional T cells (Fig S6A, cluster 4), cluster 3 still mostly maintained higher levels of these Treg signature genes, including 10-fold higher *Foxp3* expression level (Fig 4C, right). Recent studies have identified CD25 down-regulation as a key event in Tfr differentiation (Botta et al, 2017; Wing et al, 2017). Similarly, we noticed that CD25 expression is inversely correlated with the levels of CXCR5 and PD-1 in Foxp3$^+$ cells, consistent with a previous report by Wing et al (2017) (Fig S7A and B). Congruent with CD25 protein expression levels, *Il2ra* gene expression was significantly dampened in Tfr cells in cluster 3 compared with those in clusters 1 and 2 (Fig 4D). Interestingly, cluster 2 cells from ICOS FC mice displayed significantly higher *Il2ra* levels, suggesting that ICOS could be involved in CD25 down-regulation. ICOS deletion also resulted in significantly elevated CD25 protein expression in certain Foxp3$^+$ subsets (Fig S7B).

Another key event in the Treg-to-Tfr transition is CXCR5 up-regulation (Chung et al, 2011; Linterman et al, 2011; Wollenberg

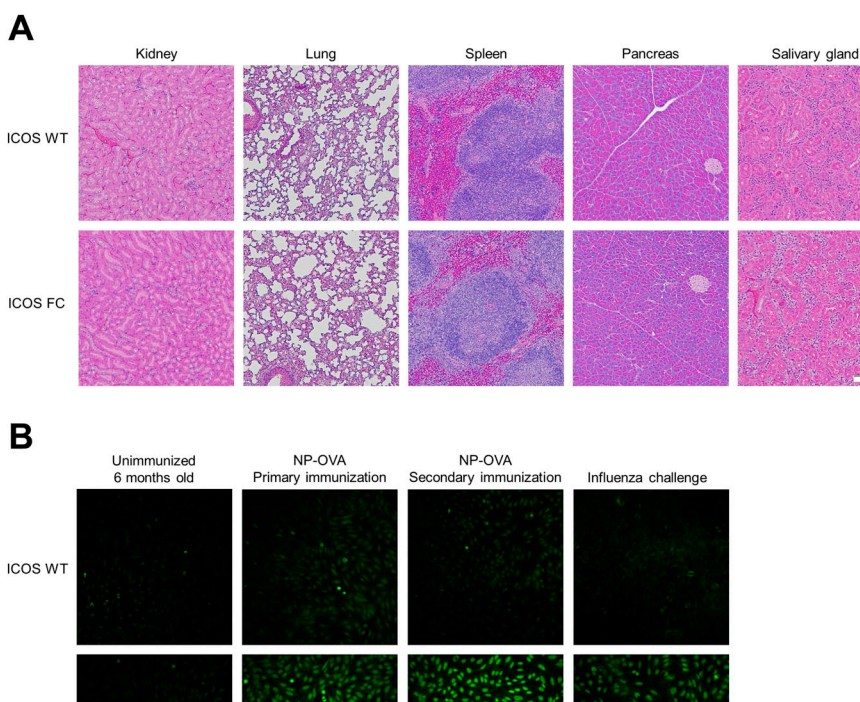

**Figure 3. ICOS FC mice produce increased anti-nuclear antibodies after immune challenge.**
**(A)** Representative images of H&E-stained tissue sections from 5-mo-old ICOS WT (n = 3) and ICOS FC (n = 3) mice. Scale bar, 200 μM. **(B)** Anti-nuclear antibodies were detected by staining HEp-2 slides with serum samples collected from mice after the following treatments. Unimmunized ICOS WT (n = 4) and ICOS FC (n = 6) mice at 6 mo of age. Primary immunization of ICOS WT (n = 5) and ICOS FC (n = 5) mice with NP-OVA/alum (serum harvested 44 dpi). Secondary NP-OVA/alum injection 30 d after the primary injection (serum harvested 44 d post-secondary challenge). IAV infection of ICOS WT (n = 5) and ICOS FC (n = 5) mice (serum harvested 30 dpi). All the samples were diluted 1:10. Scale bar, 50 μM. Nuclear fluorescence intensity was quantified using ImageJ. Data are shown as the mean ± SEM, *P < 0.05. All data are representative of two independent experiments.

et al, 2011; Vaeth et al, 2014). ICOS was previously shown to regulate CXCR5 by suppressing KLF2 expression in Tfh cells (Weber et al, 2015). This mechanism may operate in Treg/Tfr cells with high *Icos* gene expression (Fig 4E). Indeed, we observed that the gene expression of *Icos* and *Klf2* is inversely correlated and that all three clusters had higher *Klf2* expression in ICOS FC mice (Fig 4F). KLF2 was also shown to dampen Tfh differentiation by increasing S1PR1 and BLIMP-1 expression levels (Lee et al, 2015). We observed matching expression patterns of *Klf2, S1pr1,* and *Prdm1* with opposed *Bcl6* expression in all clusters (Fig 4G). Furthermore, ICOS FC mice showed an accumulation of *Klf2⁺ S1pr1⁺ Prdm1⁺* cells in cluster 2 with reduced *Bcl6* expression in clusters 2 and 3. Thus, these results suggest that ICOS is required for a few key steps in the Treg-to-Tfr transition and that failure of these processes seems to lead to an accumulation of putative Tfr precursors.

### ICOS ablation causes decreased NFAT2 activation and impaired CXCR5 expression in Tregs

To test the potential role of ICOS in up-regulating CXCR5 expression, we further analyzed our single-cell transcriptome data. We found

that cluster 2 cells from ICOS FC mice had lower *Cxcr5* expression and a trend of reduced proportion of *Cxcr5⁺* cells compared with ICOS WT mice (Fig 5A). A similar pattern was observed in cluster 1 but not in cluster 3. Next, we investigated the potential impacts of ICOS signaling on NFAT2 (product of *Nfatc1* gene), a transcription factor known to directly bind to the promoter region of C*xcr5* (Vaeth et al, 2014). Importantly, we and others have previously shown that ICOS can potentiate TCR-mediated calcium flux, a key factor in NFAT activation (Hogan et al, 2003; Leconte et al, 2016). Consistently, we found that the average expression levels of known NFAT target genes (Table S1) were significantly decreased in ICOS FC cluster 2 cells when compared to ICOS WT control (Fig 5B, top). A similar trend was seen in cluster 1 cells but not in cluster 3 cells, suggesting a link between NFAT activity and *Cxcr5* expression. Because Tfr precursor-like cells in cluster 2 also express the highest levels of *Nfatc1* (Fig 4C), we tested whether *Nfatc1* expression level was reduced in ICOS FC cluster 2 cells. However, we did not find significant differences in the expression level of the *Nfatc1* gene itself (Fig 5B, bottom). Nonetheless, we noticed that the protein levels of ICOS and NFAT2 trended higher in Tfr cells compared with Treg cells, suggesting a potential role of ICOS-NFAT2 in the Treg-to-Tfr transition (Fig 5C).

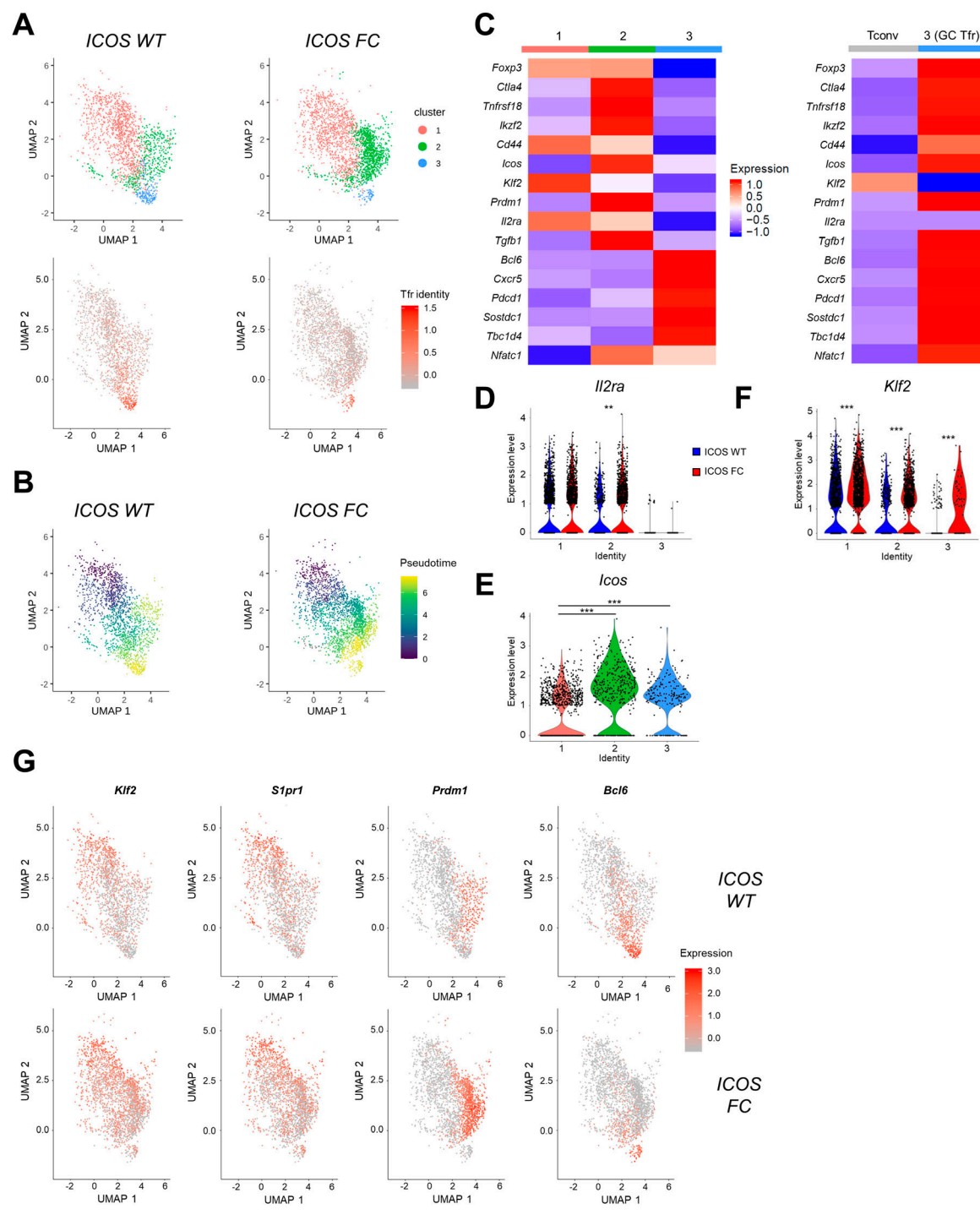

**Figure 4. ICOS-deficient Treg cells show impaired Treg-to-Tfr transition.**
Single-cell transcriptomes of FACS-sorted CD4⁺ Foxp3⁺ splenocytes from an ICOS WT or ICOS FC mouse harvested 6 d after protein immunization. **(A)** Selection and subclustering of Foxp3⁺ cells based on positive Tfr identity scores. **(B)** Pseudotime analysis showing the differentiation trajectory of selected Foxp3⁺ splenocytes. **(C)** Mean expression of regulatory and follicular genes by the indicated subpopulation. **(A, D, E, F)** *Il2ra*, (E) *Icos*, and (F) *Klf2* violin plots showing the gene expression levels subdivided by cluster identities defined in (A). **(G)** Feature plots of *Klf2*, *S1pr1*, *Prdm1*, and *Bcl6* expression. Each dot represents one cell. *$P < 0.05$, **$P < 0.01$, and ***$P < 0.001$.

Consistent with our single-cell transcriptomics data, we found that both PD-1⁻ and PD-1⁺ Tfr subsets from IAV-infected ICOS FC mice displayed significantly decreased CXCR5 expression (Fig 5D). To exclude the potential contribution of Treg cells (CXCR5⁻) in our conventional Tfr gating, we used an alternate strategy for Tfr (CD4⁺ Foxp3⁺ PD-1⁺ Bcl6⁺) and observed a similar decrease of CXCR5 expression levels in Tfr cells from ICOS FC mice (Fig S8A and B). Congruent with the unaltered *Nfatc1* mRNA levels in ICOS FC Tfr

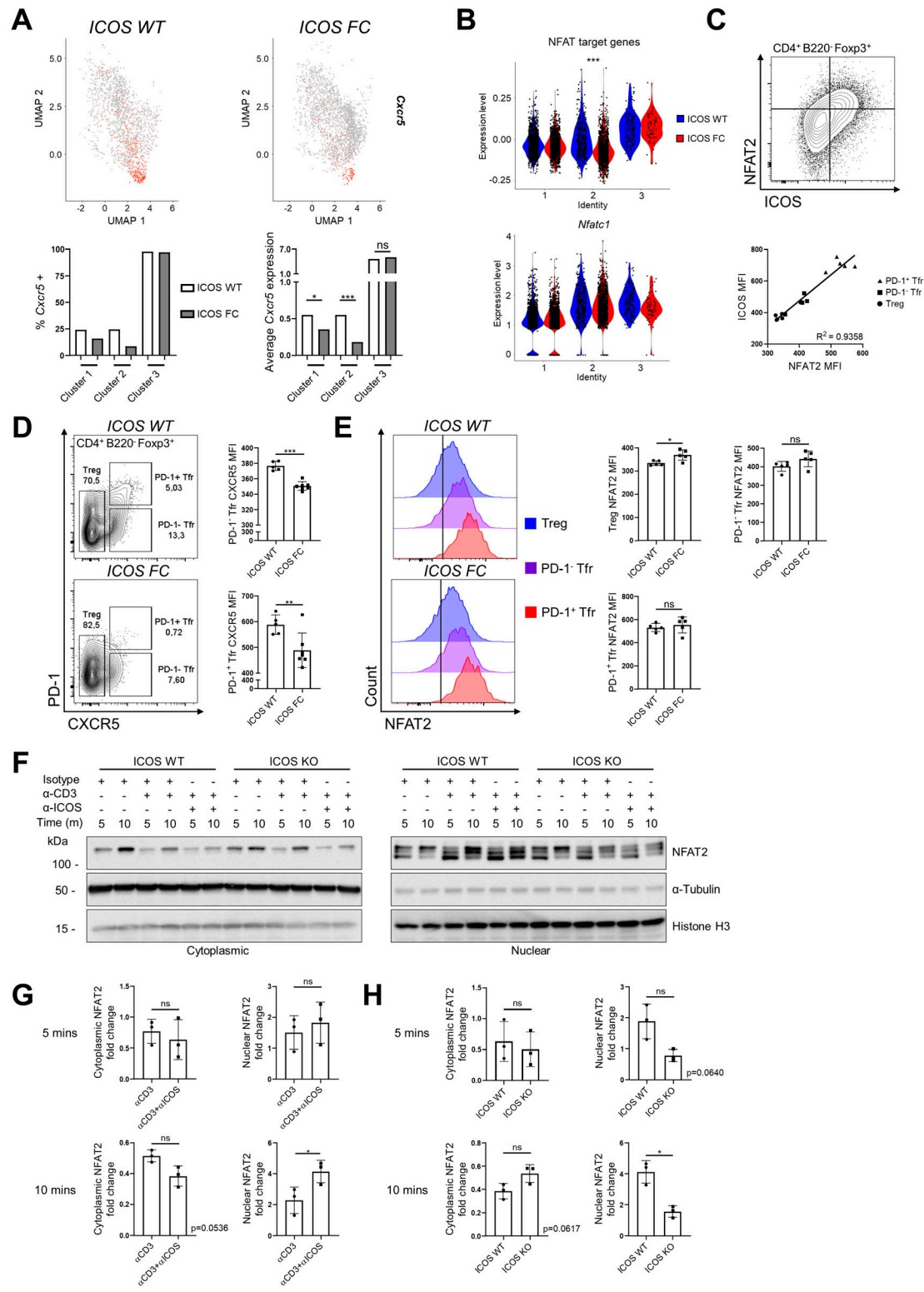

**Figure 5.  ICOS-NFAT2 signaling may regulate CXCR5 expression in regulatory T cells.**
**(A)** Feature plot of *Cxcr5* gene expression (top) and quantification of *Cxcr5*+ cells along with average *Cxcr5* expression in cluster 1, 2, and 3 cells (bottom). **(B)** Average expression of NFAT target genes (top) and expression of *Nfatc1* (bottom) subdivided by cluster identity. **(C)** Splenocytes from ICOS WT (n = 5) and ICOS FC (n = 7) mice were analyzed 30 dpi with IAV. The expression of ICOS and NFAT2 was compared in Treg, PD-1⁻ Tfr, and PD-1⁺ Tfr cells coming from ICOS WT mice. **(D)** MFI values for CXCR5 in PD-1⁻ Tfr and PD-1⁺ Tfr cells from the same splenocyte pool. **(E)** NFAT2 MFI values in ICOS WT versus ICOS FC Treg, PD-1⁻ Tfr, and PD-1⁺ Tfr cells were measured using the same splenocyte pool. **(F)** ICOS WT and ICOS germline KO CD4⁺ T cells were isolated from spleens by magnetic sorting and cultured for 2 d with α-CD3/CD28 stimulation. Cells

populations, we found that NFAT2 protein expression levels were not decreased in ICOS-deficient Tfr subsets (Fig 5E).

Because we observed that ICOS deletion led to decreased expression of NFAT target genes but not NFAT2 itself, we investigated other regulatory mechanisms. It has been established that NFAT2 activity depends on its dynamic nuclear–cytoplasmic shuttling controlled by its phosphorylation status (Hogan et al, 2003). To evaluate whether ICOS can regulate this process, we expanded purified splenic CD4+ T cells in vitro and acutely restimulated them through the TCR (CD3) with or without ICOS costimulation. Cytoplasmic and nuclear NFAT2 proteins of varying phosphorylation status were then harvested through subcellular fractionation and quantified by Western blot (Fig 5F). Without restimulation, 5- to 10-min incubation at 37°C led to an increase in hyperphosphorylated (slower migrating) NFAT2 species in the cytoplasm with concomitant disappearance of hypophosphorylated (faster migrating) NFAT2 species in the nucleus (lanes 1 and 2 in cytoplasmic and nuclear fractions). This is presumably due to temperature-induced shifts in the activity of NFAT phosphatases and kinases (Hogan et al, 2003). As predicted, CD3 ligation increased NFAT2 nuclear levels (lanes 3 and 4). Importantly, combined CD3/ICOS stimulation significantly augmented nuclear NFAT2 levels at 10 min when compared to CD3 stimulation alone (Fig 5G). This increase was dependent on ICOS because it was abolished when ICOS KO CD4+ T cells were used (Fig 5H and 10-min timepoint). Taken together, our biochemical data show that ICOS costimulation can potentiate TCR-driven NFAT2 activation in CD4+ T cells. Thus, we propose that ICOS could act upstream of the NFAT2-CXCR5 signaling axis known to be one of the key mechanisms for early Tfr differentiation (Vaeth et al, 2014).

## Discussion

In this study, we identified ICOS as a critical costimulatory receptor for Tfr differentiation upon immune challenges. We show that Foxp3-specific ICOS ablation results in altered gene expression patterns at the single-cell level leading to an accumulation of Tfr precursor-like cells and a substantial reduction in the fully differentiated "GC Tfr" population. Mice with impaired Tfr differentiation showed an increase in extraneous GC B cells without increases in total GC B-cell or Tfh cell numbers. ICOS FC mice also showed an elevated incidence of autoantibody production after GC reactions, presumably because of an expansion of autoreactive B cells. In contrast, total and virus-specific IgG2b antibody titers under steady-state and upon influenza infection (but not upon protein immunization) were diminished in ICOS FC mice. Our single-cell data strongly suggest that ICOS-mediated down-regulation of KLF2 plays a key role in shaping chemokine receptor expression and

balancing BLIMP-1-Bcl6 levels in developing Tfr cells. Our biochemical analysis demonstrates that ICOS signaling can also augment NFAT2 nuclear localization, potentially counterbalancing the negative impacts of BLIMP-1 on CXCR5 expression (Oestreich et al, 2012; Vaeth et al, 2014).

Our single-cell RNA transcriptome analysis of Foxp3+ cells from ICOS WT and ICOS FC mice indicates that dynamic changes in gene expression patterns drive the Treg-to-Tfr transition. Trajectory analysis reveals that the CD25+ BLIMP-1+–activated Treg subset progressively gains follicular features such as CXCR5 and Bcl6. Both single-cell transcriptomics and flow cytometry data demonstrate that Bcl6 and CXCR5 levels are highest in CD25− "GC-Tfr" cells, the main Foxp3+ cells shown to be found within the GC (Wing et al, 2017). During the Treg-to-Tfr transition, ICOS seems to use a mechanism that has been shown to be critical for the differentiation and maintenance of Tfh cells—timely down-regulation of KLF2 (Lee et al, 2015). As such, Tfr precursor cells reduced levels of *Klf2* and its main target genes *S1pr1* and *Prdm1* along the predicted Tfr trajectory. In contrast, *Cxcr5* and *Bcl6* expression levels were progressively elevated in cells that have dampened *Klf2* target genes, presumably because of the lack of BLIMP-1–mediated suppression of Bcl6. Importantly, the progression towards "GC-Tfr" was halted at the *Klf2*[hi] *Prdm1*[hi] stage in ICOS-deficient Treg cells. Of note, we added back-sorted Foxp3− conventional CD4+ cells as reference to our Treg cells (1:10 ratio). A small number of Tfh cells (~2%) within the add-back population may have similar changes in gene expression profile associated with KLF2 down-regulation during Tcon-Tfh conversion. However, our analysis should not have been highly affected by the Tfh cells because Tfh-lineage cells are extremely rare (~0.2% of the total cells) in our dataset. Taken together, we propose that ICOS-mediated KLF2 down-regulation is a key molecular event that initiates follicular T-cell programming in Tfr cells.

In addition to KLF2, ICOS may use NFAT2-dependent pathways to support CXCR5 expression. Although NFAT2 is highly expressed in both Tfh and Tfr cells, abrogation of NFAT2 expression in T cells was shown to cause more pronounced defects in Tfr generation as opposed to Tfh differentiation because of compromised CXCR5 expression (Vaeth et al, 2014). This observation fits well with the idea that developing Tfr cells need higher concentrations of nuclear NFAT2 to overcome elevated levels of BLIMP-1 (known repressor of the *Cxcr5* gene [Oestreich et al, 2012]) that are present in Tfr precursor cells. Our biochemical data indicate that ICOS ligation augments the amount of nuclear NFAT2 in TCR-activated CD4+ T cells. Based on these, we speculate that ICOS costimulation reinforces CXCR5 expression in early Tfr populations leading to the establishment of "GC-Tfr" differentiation.

Although we showed that ICOS signaling can maintain NFAT2 in the nucleus, the mechanism remains unclear. Nuclear transport of

were then restimulated for the indicated times with combinations of isotype control, α-CD3, and α-ICOS antibodies after which cytoplasmic and nuclear fractions were extracted and analyzed for phospho-NFAT2 by Western blot. **(G)** Bar graph representing normalized cytoplasmic versus nuclear NFAT2 fold change over isotype control of the indicated samples at the 5- and 10-min timepoints. Only the bottom NFAT2 band was quantified because it was deemed the most reliable representative of nuclear NFAT2. **(H)** Normalized nuclear NFAT2 fold change of α-CD3/α-ICOS stimulated samples from ICOS WT versus KO cells at 5 and 10 min. Data are shown as the mean ± SEM, *P < 0.05, **P < 0.01, and ***P < 0.001. All data are representative of three independent experiments.
Source data are available for this figure.

NFAT family members occurs through their dephosphorylation by calcineurin, a $Ca^{2+}$-dependent phosphatase (Hogan et al, 2003). We have previously shown that ICOS signaling can potentiate TCR-induced intracellular $Ca^{2+}$ flux, although we did not determine whether this resulted in increased NFAT activity (Leconte et al, 2016). Conversely, nuclear export of NFAT2 is triggered by phosphorylation through several kinases including GSK3$\beta$ (Beals et al, 1997). In turn, GSK3$\beta$ activity can be inhibited by Akt-mediated phosphorylation of the residue Ser9 (Yoeli-Lerner et al, 2009). We and others have shown that ICOS stimulation can increase PI3K/Akt signaling, specifically through its $Y^{181}$MFM cytoplasmic tail motif (Arimura et al, 2002; Parry et al, 2003; Gigoux et al, 2009). Thus, we suggest that ICOS could maintain NFAT2 nuclear localization by increasing its import through enhanced $Ca^{2+}$ signaling and/or decreasing its export by inhibiting GSK3$\beta$.

The biological roles of Tfr cells during GC reactions remain ill-defined. Tfr depletion studies using Bcl6 FC mice have shown decreases in antibodies specific for the immunizing antigen (Wu et al, 2016; Xie et al, 2019, 2020; Lu et al, 2021). One study also showed increases in autoantibodies and multi-organ lymphocytic infiltration in aged mice (Fu et al, 2018). Along the same line, ANAs were generated in Bcl6 FC mice after influenza infection (Botta et al, 2017; Lu et al, 2021). ICOS FC mice do not display age-related autoantibodies or lymphocytic infiltrations, suggesting that the impact of ICOS deficiency on Treg cells is weaker than that of Bcl6 deficiency. In this context, Bcl6-deficient Treg cells (as opposed to bona fide Tfr cells) could have contributed to some of these phenotypes considering recent reports that Bcl6-deficient Tregs have compromised suppressive functions in other immune settings (Sawant et al, 2012, 2015; Li et al, 2020). ICOS deficiency was shown to impair the suppressive ability of Tregs in asthma and type 1 diabetes murine models, but no defects in GC reactions were reported (Busse et al, 2012; Kornete et al, 2012). Combined with a ~fourfold reduction in the Tfr number, it seems likely that humoral immune defects in ICOS FC mice are mainly due to reduced Tfr numbers. Another potential role of Tfr cells is to promote the generation of antigen-specific antibodies. Bcl6 FC mice produce reduced amounts of IgE, IgG1, and IgG2a after immune challenges (Xie et al, 2019, 2020; Lu et al, 2021). Congruently, we found that both basal IgG2b (in unimmunized mice) and anti-influenza IgG2b titers are lower in ICOS FC mice. However, we did not find significant differences in anti-NP IgG2b titers after NP-OVA/alum immunization. Furthermore, baseline and anti-influenza IgG2c levels were normal in ICOS FC mice. These data suggest that ICOS-expressing Tfr cells may provide "helper" function to certain antibody isotypes in a context-dependent manner. Lastly, it remains possible that the affinity of the antibodies produced in ICOS FC mice may differ. Further work is required to clarify these issues.

In sum, we showed that ICOS is critically important for Tfr differentiation. ICOS-mediated down-regulation of KLF2 and its target genes can shape the Bcl6-driven Tfr programming, whereas an ICOS-NFAT2-CXCR5 signaling axis may reinforce CXCR5 expression during Tfr differentiation. Our data support the view that the main role of Tfr cells is to suppress the expansion of self-reactive GC B cells during GC reactions, and we believe that our ICOS FC mouse provides a complementary model to dissect Tfr differentiation and function.

# Materials and Methods

## Mice and animal procedures

C57BL/6 and *Foxp3*$^{YFP-Cre}$ mice (Jax 016959) (Rubtsov et al, 2008) were purchased from the Jackson Laboratory. ICOS conditional knockout mice were generated in C57BL/6 background as previously described (Panneton et al, 2018). *Foxp3*$^{YFP-Cre}$ mice were bred to generate control *Icos*$^{+/+}$; *Foxp3*$^{YFP-Cre/y}$ (ICOS WT) or *Icos*$^{fl/fl}$; *Foxp3*$^{YFP-Cre/y}$ (ICOS FC) mice. ICOS germline knockout mice have been back-crossed onto C57BL/6 background for more than 10 generations (Tafuri et al, 2001). All mice were housed in the Institut de Recherches Cliniques de Montréal animal care facility under specific pathogen-free conditions. Animal experiments were performed in accordance with animal use protocols approved by the Institut de Recherches Cliniques de Montréal Animal Care Committee. We used 8–12-wk-old male mice for experiments involving Foxp3-Cre–mediated gene deletion unless specified otherwise. For protein immunization, mice were injected intraperitoneally with 100 μg of 4-hydroxy-3-nitrophenylacetyl hapten-17 (NP17)-OVA (1 μg/ml; Biosearch Technologies) mixed with Imject Alum (Thermo Fisher Scientific) in a 1:1 ratio. For viral infections, mice were infected intranasally with a sublethal dose of IAV H1N1 (strain A/Puerto Rico/8/34 [PR8], 10 PFU/20 g body weight).

## Flow cytometry

For analysis, single-cell suspensions were prepared by mechanical disruption of spleens unless specified otherwise. Viability was assessed by staining $1 \times 10^8$ cells/ml with fixable viability dye eFluor 780 (Thermo Fisher Scientific) for 20 min at 4°C. Fc receptors were blocked using anti-CD16/CD32 (BioXCell). For intracellular staining, cells were fixed and permeabilized using the Transcription Factor Staining Buffer Set (Thermo Fisher Scientific). Surface or intracellular staining was performed at $1 \times 10^8$ cells/ml for 20 min at 4°C. The following antibodies were used: anti-CD4 BUV395 (GK1.5), anti-PD-1 BV421 (RMP1-30), anti-CD95 PE-Cy7 (Jo2), and anti-Bcl6 PE (K112-91) (BD Biosciences); anti-B220 PerCP-eFluor 710 (RA3-6B2), anti-CXCR5 biotin (SPRCL5), Streptavidin PE-Cy7, anti-ICOS FITC (7E.17G9), anti-Foxp3 APC (FJK-16s), anti-B220 eFluor 450 (RA3-6B2), and anti-CD25 PE (PC61.5) (Thermo Fisher Scientific); and anti-NFATc1 PE (7A6) and anti-GL7 FITC (GL7) (BioLegend). To identify influenza-specific B cells, we used tetramerized recombinant nucleoproteins conjugated with APC or PE (Flu tetramer) provided by Dr. Troy Randall (Allie et al, 2019). Data were acquired using a BD LSRFortessa and analyzed using FlowJo v10 (BD Biosciences).

## ELISA

Serum samples were obtained from blood collected from the submandibular vein at the indicated timepoints. Plates were coated with either goat anti-mouse IgG (SouthernBiotech), NP30-BSA and NP7-BSA (Biosearch Technologies), or heat-inactivated IAVs overnight at 4°C. Serum samples underwent twofold serial dilutions starting from the indicated initial dilution. Bound antibodies were detected using alkaline phosphatase–conjugated anti-IgG1/2b/2c/3, IgM or IgA, and *p*-nitrophenyl phosphate substrate (SouthernBiotech). The

reaction was stopped by adding 1.5 N NaOH solution, and optical density was measured at 405 nm.

## Histology

Organs were dissected and fixed in 10% neutral buffered formalin for 12 h at 4°C. Organs were then washed in 1× PBS, embedded in paraffin, and cut into 5-μM sections. Slides were stained with H&E to examine immune cell infiltration of organs. For immunofluorescence staining of Tfr cells, spleens were fixed for 2 h at 4°C in 2 ml of 4% paraformaldehyde (MilliporeSigma) followed by overnight incubation at 4°C in 2 ml of 30% sucrose (MilliporeSigma). Next, five consecutive 15-min washes at 4°C in 2 liters of 30% sucrose were performed. Sucrose was washed out, and spleens were frozen in O.C.T. medium (Tissue-Tek). Then, 10-μM sections were cut and permeabilized for 60 min at RT with 2% Triton X in PBS (MilliporeSigma). Slides were stained at 4°C overnight with a cocktail of anti-mouse CD4 PE (RM4-5), anti-mouse IgD eFluor 450 (11-26), anti-mouse GL7 Alexa Fluor 488 (GL-7), and anti-mouse Foxp3 APC (FJK-16s) (Thermo Fisher Scientific). Fluorescent signals were visualized using a DM6000 fluorescence microscope (Leica).

## Anti-nuclear antibody assay

Serum samples were obtained from blood collected from the submandibular vein at the indicated timepoints and incubated on Kallestad HEp-2 slides (Bio-Rad) according to the manufacturer's instructions. Bound antibodies were detected using goat anti-mouse IgG Alexa Fluor 555 (Thermo Fisher Scientific). Fluorophore signals were visualized using a DMRB fluorescence microscope (Leica). We recorded the mean fluorescence intensity of 10 representative nuclei per slide using the "measure" function in ImageJ. Data are presented as the mean ± SEM for each condition.

## Single-cell RNA sequencing

Splenocytes from $Foxp3^{YFP-cre}Icos^{+/+}$ (ICOS WT) and $Foxp3^{YFP-cre}Icos^{fl/fl}$ (ICOS FC) male mice were isolated 6 dpi with NP-OVA/alum and stained with anti-CD4 Alexa Fluor 647 (GK1.5; BioLegend), anti-TCRβ PE-Cy7 (H57-597; BioLegend), and propidium iodide (Thermo Fisher Scientific). Live (PI⁻), conventional (YFP⁻), and regulatory (YFP⁺) CD4⁺TCRβ⁺ T cells were sorted with a BD FACSAria (BD Biosciences) to >95% purity. Sorted conventional and regulatory T cells were mixed in a 1:10 ratio to provide an internal control. A total of 13,500 cells from ICOS WT and ICOS FC mice were sent for library preparation. Libraries were generated using the following components from 10×Genomics: Chromium Next GEM Chip G Single Cell Kit, Chromium Next GEM Single Cell 3′ GEM, Library & Gel Bead Kit v3.1, and Chromium i7 Multiplex Kit. Sequencing was performed by Genome Québec using a NovaSeq 6000 (Illumina) with a flow cell S1 PE28*91.

### Read alignment

Using Cellranger 4.0.0 (from 10×Genomics), we generated a custom reference genome using the GRCm38.p6 (mm10) assembly procured from Ensembl to which we added the Ires-Yfp-iCre sequence as described in its design map (Rubtsov et al, 2008). The alignment of

the reads was performed using the same software, and the resulting expression matrix was loaded into R, version 3.6.1 (from the R Foundation for Statistical Computing), to conduct analysis.

### Single-cell expression matrix analysis

The expression matrices were stored in an R Seurat object available in the package Seurat, version 3.0 (Stuart et al, 2019), to ease the analysis. ICOS WT and ICOS FC samples were merged during the filtering phase, which consisted of the elimination of any cell that presented more than 10% mitochondrial RNA contamination and of any cell with less than 200 unique genes expressed. The expression matrix was then log-normalized and scaled. We identified the most differentially expressed genes within the samples and proceeded with a dimensional reduction using a principal component analysis approach based on the 2,000 most variable features. We selected the first 30 most important eigenvectors produced by the principal component analysis to construct a Shared Nearest Neighbor graph and used Modularity Optimizer, version 1.3.0 (Waltman & Jan van Eck, 2013 Preprint), to identify 13 clusters. The cells were projected on a 2D space using a Uniform Manifold Approximation and Projection (UMAP) method (McInnes et al, 2020 Preprint). We isolated three clusters of interest based on their markers and moved the normalized expression matrix into an R cell_data_set object available in the package Monocle3, version 0.2.3 (Trapnell et al, 2014). Using the dimensionally reduced matrices of expression, a differentiation trajectory was constructed. The cells were then ordered along the trajectory, and pseudotime was computed. We further confirmed the consistency of our trajectory analysis using a diffusion map–based approach (available in package destiny, version 2.0.4), which has proven to be more robust to noise (Angerer et al, 2016). We computed NFAT signaling gene expression score using the AddModuleScore function available in the R library Seurat v3.0. The list of NFAT target genes was established using the PANTHER classification system combined with data from literature and can be found in Table S1 (Hermann-Kleiter & Baier, 2010; Vaeth et al, 2014; Mognol et al, 2016; Mi et al, 2021).

## CD4⁺ T-cell activation and Western blot analysis

CD4⁺ T cells were isolated from spleens and lymph nodes using the EasySep Mouse CD4⁺ T cell isolation kit (StemCell Technologies) according to the manufacturer's instructions. Purified T cells were stimulated for 2 d in complete RPMI 1640 (10% FBS, 1 U/ml penicillin, 1 μg/ml streptomycin, 55 mM β-mercaptoethanol, and 10 mM Hepes) with plate-bound anti-CD3 (3 μg/ml; BioXCell) and soluble anti-CD28 (2 μg/ml; Thermo Fisher Scientific). For restimulation, CD4⁺ T-cell blasts were incubated for 3 min at room temperature with the indicated combination of the following antibodies: 1 μg/ml Armenian hamster IgG isotype control (BioXCell), 1 μg/ml anti-CD3e (145-2C11; Thermo Fisher Scientific), and 2 μg/ml anti-ICOS (C398.4a; BioLegend). Goat anti-hamster IgG (20 μg/ml; SouthernBiotech) was added for crosslinking, and cells were immediately incubated for the indicated timepoints in a 37°C water bath. Restimulation was stopped using ice-cold STOP buffer (PBS, 10% FBS, 1 mM Na₃VO₄, and 1 mM EDTA). Cytoplasmic and nuclear fractions of restimulated T cells were obtained using the NE-PER kit (Thermo Fisher Scientific) according to the manufacturer's instructions. Lysates were boiled in

Laemmli buffer, and samples were run on SDS–PAGE. Proteins were transferred to Amersham nitrocellulose membranes (GE Health-care). Membranes were blocked using 3% BSA in TBS-T. The following antibodies were used for detection according to the manufacturer's instructions: anti-NFAT2 (D15F1; Cell Signaling Technology), anti-α-tubulin (2144; New England Biolabs), anti-histone H3 (4499; New England Biolabs), and anti-mouse IgG-HRP (Santa Cruz Biotechnology). Detection was performed using Amersham ECL prime kits (GE Healthcare), and images were captured using a ChemiDoc imaging system (Bio-Rad). Band quantification was performed using ImageJ and normalized to loading controls.

## Statistical analysis

Data are presented as the mean ± SEM unless specified otherwise. For single comparisons, statistical significance was judged using two-tailed $t$ tests. For multiple comparisons, the Holm–Sidak $t$ test was used. For single-cell gene expression comparisons between clusters, the Wilcoxon signed-rank test was used. $R^2$ values were obtained by linear regression. Statistical significance was judged based on $P$-values and is indicated as follows: *$P < 0.05$, **$P < 0.01$, and ***$P < 0.001$. Analysis was performed using Prism 7 (GraphPad Software).

# Data Availability

Single-cell transcriptome data have been deposited in the GEO database under the accession number GSE164995.

# Supplementary Information

# Acknowledgements

The authors thank Manon Laprise, Viviane Beaulieu, Stéphanie Lemay, Julie Lord, Éric Massicotte, Dominic Filion, and Simone Terouz for their technical assistance. This work was supported by the Canadian Institutes of Health Research (PJT 159526, W-K Suh; PJT 175173, JH Fritz) and the National Institutes of Health (AI153413 and AI152476, TD Randall).

## Author Contributions

V Panneton: conceptualization, data curation, formal analysis, and writing—original draft, review, and editing.
BC Mindt: data curation and writing—review and editing.
Y Bouklouch: data curation and writing—original draft, review, and editing.
A Bouchard: data curation and writing—original draft, review, and editing.
S Mohammaei: data curation and writing—review and editing.
J Chang: data curation and writing—review and editing.
N Diamantopoulos: data curation and writing—review and editing.
M Witalis: data curation and writing—review and editing.
J Li: data curation and writing—review and editing.
A Stancescu: data curation and writing—review and editing.
JE Bradley: resources and writing—review and editing.
TD Randall: resources and writing—review and editing.
JH Fritz: resources, supervision, and writing—review and editing.
W-K Suh: conceptualization, formal analysis, funding acquisition, and writing—original draft, review, and editing.

## Conflict of Interest Statement

The authors declare that they have no conflict of interest.

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
