## [Reviewer comments · Life Science Alliance]

Life Science Alliance

ICOS costimulation is indispensable for the differentiation of T follicular regulatory cells

Vincent Panneton, Barbara Mindt, Yasser Bouklouch, Antoine Bouchard, Saba Mohammaei, Jinsam Chang, Nikoletta Diamantopoulos, Mariko Witalis, Joanna Li, Albert Stancescu, John Bradley, Troy Randall, Jörg Fritz, and Woong-Kyung SUH
DOI: <https://doi.org/10.26508/lsa.202201615>

Corresponding author(s): Woong-Kyung SUH, Montreal Clinical Research Institute

Review Timeline:

Submission Date:	2022-07-19
Editorial Decision:	2022-08-15
Revision Received:	2022-12-15
Editorial Decision:	2023-01-06
Revision Received:	2023-01-19
Editorial Decision:	2023-01-20
Revision Received:	2023-01-24
Accepted:	2023-01-25

Transaction Report:

August 15, 2022

Re: Life Science Alliance manuscript #LSA-2022-01615-T

Dr Woong-Kyung SUH
IRCM
110 avenue des Pins Ouest
Montreal, Quebec H2W 1R7
Canada

Dear Dr. SUH,

Thank you for submitting your manuscript entitled "ICOS costimulation is indispensable for the differentiation of T follicular regulatory cells" to Life Science Alliance. The manuscript was assessed by expert reviewers, whose comments are appended to this letter. We invite you to submit a revised manuscript addressing the Reviewer comments.

Thank you for this interesting contribution to Life Science Alliance. We are looking forward to receiving your revised manuscript.

Sincerely,

B. MANUSCRIPT ORGANIZATION AND FORMATTING:

Reviewer #1 (Comments to the Authors (Required)):

In this manuscript by Panneton et al., the authors show that ablation of ICOS in Foxp3-expressing cells results in reduced follicular regulatory (Tfr) T cells. Ablation of ICOS also resulted in a reduction in flu-specific GC B cells and IgG1 and IgG2 along with an increase in autoreactive B cells. The reduction in Tfr cells in ICOS Fc mice might be due to increase Klf2 in Tfr-precursor cells, as indicated by single cell RNA-seq analysis.

Overall, this study represents a minor advancement in our understanding of the signals promoting Tfr cell development. It has already been well established that ICOS is required for Tfr cell development and that Tfr cells develop from Treg cells. Therefore, it is not a surprise that ablation of ICOS in Foxp3+ cells would lead to a decrease in Tfr cells. Similarly, several studies have assessed the B cell response following influenza infection in mice lacking Tfr cells and found that there was a reduced influenza-specific germinal center and antibody response in the absence of Tfr cells (Botta et al Nature Immunology 2017, Lu et al JEM 2020). These studies also showed that the absence of Tfr cells resulted in an altered BCR repertoire and the development of autoreactive B cells.

This study does have some novelty in that it is the first to specifically ablate ICOS in Foxp3+ cells. While the results of this ablation may be predictable, there is still value in their publication. Additionally, it is interesting that ablation of ICOS does not result in a decrease in the number of Treg cells. Rather, their single-cell analysis indicates that there is a build-up of Tfr cell precursor cells.

Major issues:

- 1) The manuscript does not reference Lu et al JEM 2020. This is a major oversight because several of the figures in this paper are nearly identical to results shown in that work. Therefore, this reference should be added, and the authors should discuss how their results are consistent with published work.
- 2) The authors propose that ICOS acts upstream of NFAT2 to regulate Tfr cell differentiation based on their western blot data. However, they only quantify the nuclear and cytoplasmic NFAT signal at 10 minutes for certain groups. The authors should show the cytoplasmic and nuclear NFAT2 fold change over isotype control for all groups and time points shown in their representative image as there do appear to be differences in the result based on the time point chosen.

Additionally, the authors only quantify the bottom band when assessing nuclear NFAT2. The authors should clarify why only the hypophosphorylated band is relevant for assessing nuclear NFAT2. Ideally the authors should separately quantify all 3 NFAT2 bands to assess how they change over time in the different groups.

Minor issues:

- 1) Gene names should be italicized in all figures.
- 2) Labeling the ICOS knockout mice ICOS FC is not intuitive. It would be preferable to label these mice ICOS cKO (or something similar that makes it clear that ICOS is being ablated).
- 3) Legend showing that blue is ICOS WT and red is ICOS Fc should be added to the violin plots in figure 5. This legend should also be shown the first time this color scheme is used (Fig. 4D). Currently it is not shown until the Klf2 violin plot later in the figure.
- 4) The Klf2 violin plot should be its own panel. Right now, it shares a panel with the Icos expression data.
- 5) The western blot data shown in Fig. 5F appears somewhat blurry. If possible, it would be useful to use a higher resolution image.

Reviewer #2 (Comments to the Authors (Required)):

Panneton and colleagues report on the actions of ICOS during Tfr formation using a conditional deletion model. This model potentially permits further analysis of the actions of Tfr cells during immunity, and the importance of the signals that leads to their formation. They present interesting although somewhat contradictory findings for the outcome of immunity in the absence of Tfr cells. They investigate the importance of ICOS during Tfr ontogeny, although there are potential confounding factors in their experimental approach that undercut the significance of these findings.

Major comments

1. Conditional deletion of ICOS resulted in a reduction in Tfr populations. The authors state: interestingly, these reduced Tfr populations were balanced by an increased proportion of Treg cells. Based on the gating strategies, surely this is inevitable; the relative proportions of these two populations are completely dependent. Any associated change in the Treg population would need to be determined by absolute counts, and this does not appear to be the case. Thus, the conclusion that there is an impairment in Treg to Tfr transition does not seem to be supported by the data presented.
2. After influenza infection, ICOS deletion appears to result in a reduction of flu-specific B cells. Summary plots should be provided and the authors should ensure that the flu-tetramer/B220 contour plot is representative of the results of this experiment. Also, the gating strategy is a bit unclear. The legend suggests that the plots in the left column of Fig. 2C were gated on B220, if so there is no need to show B220 again in the tetramer plots.
3. The authors state: This results in a significant increase of extraneous GC B cells, some of which could be autoreactive in nature... I don't know what this means. The statement should be clarified.
4. ICOS deficiency was also associated with a reduction in total and virus-specific IgG2b. The presentation of these results is suboptimal. We are left to assume that the serum was diluted but the dilution series should be shown on the x-axes of the plots. IgG2c responses appeared to be intact. The authors have not commented on this observation. It appears to me that it is not so much that the antibody titre was lower (read out by the dilution at which the curve reaches baseline), but that the slope of the curve changes, which is sometimes an indication of a difference in affinity. The authors should comment.
5. Next they examined the formation of Tfr after immunisation. The text states that at D6 after immunisation, Tfrs are just beginning to form. The sorting strategy is based on Foxp3 expression. According to the heatmaps in Fig. 4A, the Tfr population is Foxp3⁻. Given the relative deficiency of Tfr cells conferred by Icos deficiency, and the sorting strategy employed, there is a significant risk that the putative nascent Tfr cells are contaminated with Tfh cells. It is not clear how they have controlled for this potential confounding effect. Furthermore, cluster 5 in Fig. S4B, which they defined as Tfr-like using their own definition, appears to be abundant in ICOS FC mice, and contains very few cells that meet the Tfr identity score. What are the non-Tfr-like cells in this cluster?
6. While Fig. 4C left panel suggests cluster 3 cells are Foxp3⁻, the right panel suggests that Foxp3 is increased relative to Tconv cells. The fold change should be included.
7. A clear distinction between contaminating Tfh cells in the Tfr-like population is crucial, since the subsequent analysis depends on the phenotype of this cluster. They demonstrate a defect in Klf2 downregulation. This is reasonable since ICOS is known to be important for this step in Tfh formation but the evidence that the effect is specific for Tfr cells is not convincing. Furthermore, they postulate that Icos results in a reduction in Il2ra and Cxcr5 expression. These analyses are problematic because of the difficulty of defining the population for analysis independently of the marker of interest. Thus, Tfr cells are defined by CXCR5 expression, and then the level of CXCR5 expression is assessed. How can they be distinguished between reduced CXCR5 expression, and contamination of CXCR5⁻ (i.e. conventional Tregs) without an independent marker?
8. Finally, they show that while NFAT expression is increased in Tfr cells, they postulate altered function as a result of a change in nuclear shuttling in the absence of ICOS. Results are presented to suggest that the kinetics of NFAT2 phosphorylation is altered such that hypophosphorylated nuclear NFAT2 is increased at the 10min timepoint by ICOS ligation. Tfh cells should be included for comparison. Since the result hinges on a single time point, it would be ideal if the time course were extended to determine if the effect is on the duration of nuclear accumulation of hypophosphorylated NFAT2.

Other comments

1. In the first section of the results, there appears to be a problem with the definition of PD-1⁺ Tfr as they are said to be CXCR5⁻ in the text, but the representative FACS plot suggests that they are CXCR5⁺ (as expected).
2. They show that mice develop ANAs after immunisation or infection. The presentation of the results is unusual, with ANAs being reported as MFI. They should be reported in the conventional way according to the serial serum dilution.

Reviewer #3 (Comments to the Authors (Required)):

The authors show that ICOS deficiency in Treg-lineage cells drastically reduces the number of Tfr cells during GC reactions but has a minimal impact on conventional Treg cells by using ICOS FC mice. SC transcriptome analysis of Foxp3+ cells at an early stage of the GC reaction suggests that ICOS can inhibit Klf2 expression and promote nuclear localization of NFAT2 in turn promote Tfr differentiation. The humoral responses from these mice after immunizations are in line with previous reports.

Points to be addressed:

1. FACS of baseline Tfh and Tfr to show the any defects in ICOS FC mice. To me, the phenotypes are mainly showed up after some kind of immunological challenge.
2. Figure 1: please show titer of NP specific IgG2b antibody. It will be a good confirmatory data.
3. Figure 2: It would be interesting to check flu specific IgA titers. This paper "PMID: 26887860" actually showed elevated IgA level in the absence of Tfr cells.
4. This argument is weak. If the author wants to keep it, please move it to discussion section with more reference. "we noticed a significant decrease of Tgfb1 expression in ICOS FC cluster 3 cells (Fig. 4 E). This could explain reduced IgG2b titers observed in ICOS FC mice since TGF- β 1 is a known class switch factor for this isotype"
5. Fig 5A Could you please also show cxcr5 expression in cluster 3.
6. Supplement Fig2, please switch marker labels.

We would like to thank you for your thoughtful comments and critiques. We carefully revised our manuscript in response to your feedback. Please find below a point-by-point reply to each comment with references to the revised text (highlighted in yellow; page numbers in this reply), Figure numbers, or Supplementary Figure numbers.

Reviewer #1 (Comments to the Authors (Required)):

In this manuscript by Panneton et al., the authors show that ablation of ICOS in Foxp3-expressing cells results in reduced follicular regulatory (Tfr) T cells. Ablation of ICOS also resulted in a reduction in flu-specific GC B cells and IgG1 and IgG2 along with an increase in autoreactive B cells. The reduction in Tfr cells in ICOS Fc mice might be due to increase Klf2 in Tfr-precursor cells, as indicated by single cell RNA-seq analysis.

Overall, this study represents a minor advancement in our understanding of the signals promoting Tfr cell development. It has already been well established that ICOS is required for Tfr cell development and that Tfr cells develop from Treg cells. Therefore, it is not a surprise that ablation of ICOS in Foxp3+ cells would lead to a decrease in Tfr cells. Similarly, several studies have assessed the B cell response following influenza infection in mice lacking Tfr cells and found that there was a reduced influenza-specific germinal center and antibody response in the absence of Tfr cells (Botta et al Nature Immunology 2017, Lu et al JEM 2020). These studies also showed that the absence of Tfr cells resulted in an altered BCR repertoire and the development of autoreactive B cells.

This study does have some novelty in that it is the first to specifically ablate ICOS in Foxp3+ cells. While the results of this ablation may be predictable, there is still value in their publication. Additionally, it is interesting that ablation of ICOS does not result in a decrease in the number of Treg cells. Rather, their single-cell analysis indicates that there is a build-up of Tfr cell precursor cells.

Major issues:

1) The manuscript does not reference Lu et al JEM 2020. This is a major oversight because several of the figures in this paper are nearly identical to results shown in that work. Therefore, this reference should be added, and the authors should discuss how their results are consistent with published work.

> Thank you for the reminder. It was our oversight. We cited the reference (Lu et al JEM 2020) in several relevant sections (Page 4, 7, 8, 16, 17).

2) The authors propose that ICOS acts upstream of NFAT2 to regulate Tfr cell differentiation based on their western blot data. However, they only quantify the nuclear and cytoplasmic NFAT signal at 10 minutes for certain groups. The authors should show the cytoplasmic and nuclear NFAT2 fold change over isotype control for all groups and time points shown in their representative image as there do appear to be differences in the result based on the time point chosen.

> The amount of nuclear NFAT2 appears to be slightly elevated by ICOS costimulation both at 5 and 10 min timepoints in the representative Western blot (Fig. 5F). However, when we averaged data from

three independent experiments, statistically significant differences were observed only at 10 min post-stimulation. We now quantified 5 and 10 min timepoints for all groups and updated the results in Fig. 5G and Fig.5H. The figure legend is revised accordingly (Page 21). Thus, in our antibody-mediated T cell stimulation experiments, the 10 min timepoint seems to be the optimal condition to reveal impacts of ICOS costimulation.

Additionally, the authors only quantify the bottom band when assessing nuclear NFAT2. The authors should clarify why only the hypophosphorylated band is relevant for assessing nuclear NFAT2. Ideally the authors should separately quantify all 3 NFAT2 bands to assess how they change over time in the different groups.

> We quantified the intermediate bands. However, it was difficult to see any consistent differences between conditions. Considering that there are 13 serine residues in NFAT proteins that are known to be phosphorylated (PMID: 11402342), we presume that the intermediate bands represent numerous phospho-NFAT2 species undergoing dynamic changes. Some of the intermediate phospho-NFAT2 proteins may not be in the nucleus as we saw some intermediate bands in the cytoplasmic fractions (Fig. 5F, Cytoplasmic). In contrast, the bottom band (mostly likely fully dephosphorylated species) showed up exclusively in the nuclear fractions. With these, we believe that the bottom band is the most reliable representative of nuclear NFATs. We clarified this in the figure legend (Page 21).

Minor issues:

1) *Gene names should be italicized in all figures.*

> We made sure that all the gene names are italicized in all figures.

2) Labeling the ICOS knockout mice ICOS FC is not intuitive. It would be preferable to label these mice ICOS cKO (or something similar that makes it clear that ICOS is being ablated).

> We would like to keep the nomenclature as is. This “FC” nomenclature has been used in multiple published papers to refer to Foxp3-Cre mediated knockout mouse model “Bcl6 FC” (PMID: 26887860; PMID: 30013575; PMID: 31434804). To prevent any confusion, we defined our nomenclature in *Materials and methods* (newly added; Page 22) and in the early part of *Results* section (was in the original manuscript; Page 6).

3) *Legend showing that blue is ICOS WT and red is ICOS Fc should be added to the violin plots in figure 5. This legend should also be shown the first time this color scheme is used (Fig. 4D). Currently it is not shown until the Klf2 violin plot later in the figure.*

> Good point. We updated legends for Figures 4D and 5B accordingly.

4) *The Klf2 violin plot should be its own panel. Right now, it shares a panel with the Icos expression data.*

> The Klf2 violin plot is now in Fig. 4F.

5) The western blot data shown in Fig. 5F appears somewhat blurry. If possible, it would be useful to use a higher resolution image.

> We re-exported the image at a higher resolution and revised Fig. 5F. In addition, we will deposit the source images along with the figure files and work with the staff to ensure the quality of the figures.

Reviewer #2 (Comments to the Authors (Required)):

Panneton and colleagues report the on the actions of ICOS during Tfr formation using a conditional deletion model. This model potentially permits further analysis of the actions of Tfr cells during immunity, and the importance of the signals that leads to their formation. They present interesting although somewhat contradictory findings for the outcome of immunity in the absence of Tfr cells. They investigate the importance of ICOS during Tfr ontogeny, although there are potential confounding factors in their experimental approach that undercut the significance if these findings.

Major comments

1. Conditional deletion of ICOS resulted in a reduction in Tfr populations. The authors state: interestingly, these reduced Tfr populations were balanced by an increased proportion of Treg cells. Based on the gating strategies, surely this is inevitable; the relative proportions of these two populations are completely dependent. Any associated change in the Treg population would need to be determined by absolute counts, and this does not appear to be the case. Thus, the conclusion that there is an impairment in Treg to Tfr transition does not seem to be supported by the data presented.

> We agree with the reviewer that the changes in the absolute Treg counts are critical to support our argument in this section. We revised the relevant text in order to avoid overinterpretation of our results (Page 6-7).

2. After influenza infection, ICOS deletion appears to result in a reduction of flu-specific B cells. Summary plots should be provided and the authors should ensure that the flu-tetramer/B220 contour plot is representative of the results of this experiment. Also, the gating strategy is a bit unclear. The legend suggests that the plots in the left column of Fig. 2C were gated on B220, If so there is no need to show B220 again in the tetramer plots.

> We confirm that the flu tetramer plots are representative of the results of this experiment. The summary plots are on the right side of panel C. To clarify our results, we replaced the flu-tetramer/B220 contour plot in Fig. 2C with a histogram of the flu-tetramer signal in B220⁺GL7⁺Fas⁺ GC B cells.

3. The authors state

This results in a significant increase of extraneous GC B cells, some of which could be autoreactive in nature...

I don't know what this means. The statement should be clarified.

> We rephrased this statement as follows: “This results in a significant increase of non-IAV specific GC B cells, some of which could be autoreactive in nature.” (Page 7).

4. ICOS deficiency was also associated with a reduction in total and virus-specific IgG2b. The presentation of these results is suboptimal. We are left to assume that the serum was diluted but the dilution series should be shown on the x-axes of the plots. IgG2c responses appeared to be intact. The authors have not commented on this observation. It appears to me that it is not so much that the antibody titre was lower (read out by the dilution at which the curve reaches baseline), but that the slope of the curve changes, which is sometimes an indication of a difference in affinity. The authors should comment.

> We added the following sentence in the relevant figure legends (Page 19): “All the serum samples underwent two-fold serial dilutions starting from...”. With this clarification, we would like to leave the X-axis as is to avoid overcrowding. Regarding the intact anti-viral IgG2c response and the possibility of altered affinity, we dedicated a section in *Discussion* (Page 17).

5. Next they examined the formation of Tfr after immunisation. The text states that at D6 after immunisation, Tfrs are just beginning to form. The sorting strategy is based on Foxp3 expression. According to the heatmaps in Fig. 4A, the Tfr population is Foxp3⁻. Given the relative deficiency of Tfr cells conferred by Icos deficiency, and the sorting strategy employed, there is a significant risk that the putative nascent Tfr cells are contaminated with Tfh cells. It is not clear how they have controlled for this potential confounding effect.

> We agree with the reviewer that contamination of the Tfr clusters with Tfh cells is possible. However, we argue that this should not be a significant issue.

We have performed flow cytometry analyses of splenocytes 6 days post-immunization with NP-OVA/alum (the same conditions used in our single cell RNA sequencing experiments) to estimate Tfh/Tfr populations (new Supplementary Figure 5). Under this condition, Tfh cells represent 2 % of CD4⁺Foxp3⁻ cells both in ICOS WT and FC mice (Fig. S5B).

Therefore, we expect that ~0.2 % of our single cells represent Tfh cells (2 % of the 10 % Foxp3⁻ cells we spiked in as a control). We do not believe this small portion of Tfh cells heavily skew the clustering patterns of the bulk Treg/Tfr populations. In addition, the absolute number of Tfh cells in the total sample is estimated to be ~10. If we presume most Tfh cell would be falling in the “GC Tfr” cluster (cluster 3 in Fig. 4) due to the follicular signature genes, it represents around ~5% of the total cells in cluster 3 in ICOS WT. Based on these, we judge that the contaminating Tfh cells should not severely bias our interpretation of the Tfr differentiation path. We acknowledged this caveat and rationale in *Results* (Page 9).

Furthermore, cluster 5 in Fig. S4B, which they defined as Tfr-like using their own definition, appears to be abundant in ICOS FC mice, and contains very few cells that meet the Tfr identity score. What are the non-Tfr-like cells in this cluster?

> Cluster 5 in Fig. S4B (now Fig. S6B) contains regulatory T cells with activated phenotypes, some of which start showing Tfr identity. This cluster is equivalent to cluster 2 in the analysis shown in Fig. 4.

We clarified this in the relevant text (Page 9). We believe that this cluster accumulates in ICOS FC mice due to an inefficient Treg-to-Tfr transition.

6. While Fig. 4C left panel suggests cluster 3 cells are *Foxp3*⁻, the right panel suggest that *Foxp3* is increased relative to *Tconv* cells. The fold change should be included.

> There is a 10-fold increase of *Foxp3* expression level between *Tconv* cells and cluster 3 *Tfr* cells. We added this information in the text (Page 10).

7. A clear distinction between contaminating *Tfh* cells in the *Tfr*-like population is crucial, since the subsequent analysis depends on the phenotype of this cluster. They demonstrate a defect in *Klf2* downregulation. This is reasonable since ICOS is known to be important for this step in *Tfh* formation but the evidence that the effect is specific for *Tfr* cells is not convincing.

> As detailed in response to point 5, *Tfh* cells represent only 0.2 % of the total single cells data points and it is likely that the *Tfh* cells are concentrated in cluster 3 (in Fig. 4). We found that the downregulation of *Klf2* gene expression happens gradually from cluster 1 to cluster 2 and then cluster 3 (Fig. 4C and 4G). Based on this, we believe that *Klf2* downregulation happens during Treg differentiation into *Tfr*.

Furthermore, they postulate that *Icos* results in a reduction in *Il2ra* and *Cxcr5* expression. These analyses are problematic because of the difficulty of defining the population for analysis independently of the marker of interest. Thus, *Tfr* cells are defined by *CXCR5* expression, and then the level of *CXCR5* expression is assessed. How can they distinguished between reduced *CXCR5* expression, and contamination of *CXCR5*⁻ (i.e. convention Tregs) without an independent marker?

> We agree that this is an important point. To address this issue, we have performed flow cytometry as part of influenza infection experiments using the following alternate gating strategy for *Tfr* cells: *CD4*⁺ *B220*⁻ *Foxp3*⁺ *PD-1*⁺ *Bcl6*⁺. Using this strategy, we confirmed that the average expression of *CXCR5* by *Tfr* cells is decreased in ICOS FC mice. These results are described in the text (Page 12) along with a new Supplementary Figure (Fig. S8).

For our *CD25* analysis (originally Fig. S6; now Fig. S7), we agree that the *CXCR5*^{lo} *PD-1*^{lo} gate may contain conventional Tregs. However, it is unlikely that conventional Tregs would be found in the *CXCR5*^{int} *PD-1*^{int} and *CXCR5*^{hi} *PD-1*^{hi} gates. Nevertheless, we have removed the term “*Tfr*” cells in this figure and the corresponding text. Our conclusions remain the same.

8. Finally, they show that while *NFAT* expression is increased in *Tfr* cells, they postulate altered function as a result in a change in nuclear shuttling in the absence of ICOS. Results are presented to suggest that the kinetics of *NFAT2* phosphorylation is altered such that hypophosphorylated nuclear *NFAT2* is increased at the 10min timepoint by ICOS ligation. *Tfh* cells should be included for comparison. Since the

result hinges on a single time point, it would be ideal if the time course were extended to determine is the effect is on the duration of nuclear accumulation of hypophosphorylated NFAT2.

> We need more than 24 million cells per genotype for the typical Western blot experiment shown in Fig. 5F. Because of this, we cannot use *ex vivo* Tfh or Tfr cells isolated from immunized mice. Instead, we used *in vitro* activated CD4⁺ T cells as described in the original manuscript. In our hands, the impact of ICOS ligation in NFAT nuclear translocation peaked at 10 min and started declining by 20 min. This is presumably due to internalization of cross-linked antibodies. However, these timepoints seem relevant to *in vivo* T-B interactions. *In vivo* imaging data have shown that follicular T cells form conjugates with GC B cells for 5-20 minutes (PMID: 23619696; PMID: 25317561). Further, ICOS-dependent calcium flux (the upstream signal for NFAT nuclear localization) happens transiently during this T-B contact (PMID: 25317561). With these, we would like to keep the data as is.

Other comments

1. In the first section of the results, there appears to be a problems with the definition of PD-1+ Tfr as they are said to be CXCR5- in the text, but the representative FACS plot suggests that they are CXCR5+ (as expected).

> This was an oversight; the text is now corrected (Page 6).

2. They show that mice develop ANAs after immunisation or infection. The presentation of the results is unusual, with ANAs being reported as MFI. They should be reported in the conventional way according the serial serum dilution.

> We agree that serial dilution would be ideal. However, if the sample numbers are sufficient and the contrast is strong between conditions, an alternative way is to test all the samples at a fixed dilution rate and to compare the percentages of ANA positive cases (PMID: 28892471). Indeed, when we first established ANA protocol, we compared multiple serum dilution conditions (undiluted, 1:5, and 1:10) and chose 1:10 dilution since it gave us low background and clear contrast between genotypes. Afterwards, we fixed the dilution rate at 1:10 for all the following experiments. Our MFI analysis reflects our assessment in a more quantitative manner compared to simple percentage plots. With this, we would like to add the dilution rate in the Fig. 3 legend (Page 21) and leave the data presentation as is.

Reviewer #3 (Comments to the Authors (Required)):

The authors show that ICOS deficiency in Treg-lineage cells drastically reduces the number of Tfr cells during GC reactions but has a minimal impact on conventional Treg cells by using ICOS FC mice. SC transcriptome analysis of Foxp3+ cells at an early stage of the GC reaction suggests that ICOS can inhibit Klf2 expression and promote nuclear localization of NFAT2 in turn promote Tfr differentiation. The humoral responses from these mice after immunizations are in line with previous reports.

Points to be addressed:

1. FACS of baseline Tfh and Tfr to show the any defects in ICOS FC mice. To me, the phenotypes are mainly showed up after some kind of immunological challenge.

> We performed flow cytometry analyses of splenocytes harvested from unimmunized mice. We found that the proportion of PD-1⁺ Tfr cells is reduced even in unimmunized ICOS FC mice (new Supplementary Figure, Fig. S2; text revised accordingly, Page 6). These results suggest that ICOS is involved in the generation of Tfr cells in steady-state antibody responses against environmental antigens. However, we think that the detailed analysis of this process is out of the scope of this manuscript.

2. Figure 1: please show titer of NP specific IgG2b antibody. It will be a good confirmatory data.

> We performed NP-OVA/alum immunization experiments and added the results (Anti-NP30 IgG2b and Anti-NP7 IgG2b, Fig. 1E). Unexpectedly, we did not observe any differences in IgG2b titers in ICOS FC mice upon immunization. This suggest that ICOS-expressing Tfr cells play different roles in IgG2b responses depending on immunological settings (steady state vs protein immunization with alum vs influenza infection). Certainly, we need further experiments to explain these intriguing observations. However, we feel that this is out the scope of this manuscript. We revised the text in relevant sections: *Results* (Page 6-8), *Discussion* (Page 14; Page 17), and figure legend (Page 19).

3. Figure 2: It would be interesting to check flu specific IgA titers. This paper "PMID: 26887860" actually showed elevated IgA level in the absence of Tfr cells.

> It is very difficult for us to repeat flu infection experiments to measure anti-viral IgA titers since the collaborator Barbara Mindt moved on to do her Postdoc and Dr. Jorg Fritz group is left short-handed. Instead, we tried to see if there are any elevated IgA titers in our protein immunization model. There were very low anti-NP IgA titers in both ICOS WT and ICOS FC mice in our experimental model making it difficult to get reliable ELISA readings even when we started with 1:25 dilution. However, from the readings we got there was no indication that ICOS FC mice had highly elevated IgA levels. We would like to examine anti-Flu IgA titers in ICOS FC mice if opportunities arise in the future. However, we feel it is too costly for us to do it now in the context of the current manuscript.

4. This argument is weak. If the author wants to keep it, please move it to discussion section with more reference. "we noticed a significant decrease of Tgfb1 expression in ICOS FC cluster 3 cells (Fig. 4 E). This could explain reduced IgG2b titers observed in ICOS FC mice since TGF- β 1 is a known class switch factor for this isotype"

> We agree with the reviewer. Further, measurements of anti-NP IgG2b titers performed as part of this revision did not align with findings from steady-state or viral infection experiments. Considering this, we removed the *Tgfb1* sections from *Results* and *Discussion* sections.

5. Fig 5A Could you please also show *cxcr5* expression in cluster 3.

> We added mean %*Cxcr5*⁺ cells and average *Cxcr5* expression data for cluster 3 (Fig. 5A). We noticed that *Cxcr5* RNA levels are not significantly different in cluster 3 cells between ICOS WT and FC. Thus, cluster 2 cells show clear differences between genotypes in *Cxcr5* expression levels, cluster 1 cells show

smaller differences, and cluster 3 cells show no differences. Interestingly, a similar pattern is observed in NFAT target genes supporting the potential link between NFAT activity and *Cxcr5* expression in Tfr precursor cells. The relevant text (Page 11) and figure legend (Page 20) are revised to emphasize these points.

6. *Supplement Fig2, please switch marker labels.*

> We revised the GC marker labeling scheme for clarity (Supplementary Fig. 2 in the original, Supplementary Fig. 3 after revision).

January 6, 2023

Re: Life Science Alliance manuscript #LSA-2022-01615-TR

Dr. Woong-Kyung SUH
Montreal Clinical Research Institute
110 avenue des Pins Ouest
Montreal, Quebec H2W 1R7
Canada

Dear Dr. SUH,

Thank you for submitting your revised manuscript entitled "ICOS costimulation is indispensable for the differentiation of T follicular regulatory cells" to Life Science Alliance. The manuscript has been seen by the original reviewers whose comments are appended below. While the reviewers continue to be overall positive about the work in terms of its suitability for Life Science Alliance, some important issues remain.

Our general policy is that papers are considered through only one revision cycle; however, given that the suggested changes are relatively minor, we are open to one additional short round of revision. Please note that I will expect to make a final decision without additional reviewer input upon re-submission.

Please submit the final revision within one month, along with a letter that includes a point by point response to the remaining reviewer comments.

To upload the revised version of your manuscript, please log in to your account: <https://lsa.msubmit.net/cgi-bin/main.plex>
You will be guided to complete the submission of your revised manuscript and to fill in all necessary information.

B. MANUSCRIPT ORGANIZATION AND FORMATTING:

Sincerely,

Reviewer #1 (Comments to the Authors (Required)):

The manuscript is now suitable for publication.

Reviewer #2 (Comments to the Authors (Required)):

1. I don't think the claim that there is an increase in the number of non-IAV GC B cells is justified or supported by data shown.

2. I am not satisfied with the description of the antibody titres in Fig. 2E and Fig. S4.

The statement: However, we found significant decreases of IAV-specific IgG2b serum titers (Fig. 2E).

The authors appear to conflate OD and titre. A change in viral specific antibody titres means that the dilution at which antigen-specific antibodies are detected is different. As far as I can see, the serum dilution at which antiviral antibodies is detectable is similar in WT and FC. Instead, the authors report that the OD is different between the groups at each particular serum dilution. This might indicate a difference in antibody concentration at each dilution, which could be determined using a standard curve or, as mentioned previously, it might indicate a difference in antibody affinity but proof for this conclusion would require additional investigation?

The titres should be shown.

Incidentally, a similar misunderstanding of serum titre occurs in the reporting of the ANAs in fig. 3B. ANAs are conventionally reported according to the highest serum dilution at which they remain positive. The MFI of positive cells does not correlate with serum dilution.

3. Contamination of Tfr with Tfh.

My concern was not that the results would be confounded by Tfh cells in the add-back population, rather that cells called nascent Tfr might be Tfh cells formed in the experimental group after immunisation. The authors have attempted to distinguish between Tfh and Tfr using the reference (add-back) population but without barcoding it is impossible to distinguish rare add-back Tfh and use them to determine in which subset rare true Tfh cells are located. The question is, how can we be sure that cluster 3 (Fig. 4C) are not nascent Tfh cells? This is not resolved by the comparison shown in the right panel of Fig. 4C as this is a comparison between cluster 3 and conventional T cells. The comparison should be between cluster 3 and bona fide Tfh cells.

Reviewer #3 (Comments to the Authors (Required)):

The authors addressed all my questions. I recommend accepting their work for publication.

We appreciate the positive feedback from the reviewers. As for the remaining concerns raised by reviewer #2, we revised our manuscript to enhance the clarity. Our responses are in red and the revised texts are highlighted in yellow. There is no change in Figures and Supplementary Figures.

Reviewer #1 (Comments to the Authors (Required)):

The manuscript is now suitable for publication.

Reviewer #2 (Comments to the Authors (Required)):

1. I don't think the claim that there is an increase in the number of non-IAV GC B cells is justified or supported by data shown.

> We agree that our conclusion was too strong considering that the increase in the absolute number of non-IAV GC B cells was a strong trend with an insufficient statistical significance. However, our conclusion remains the same since there was a statistically significant increase in the proportion of non-IAV GC B cells. Thus, we revised the text as follows: "These results strongly suggested an increase of non-IAV specific GC B cells, ..." (page 7).

2. I am not satisfied with the description of the antibody titres in Fig. 2E and Fig. S4.

The statement: However, we found significant decreases of IAV-specific IgG2b serum titers (Fig. 2E).

The authors appear to conflate OD and titre. A change in viral specific antibody titres means that the dilution at which antigen-specific antibodies are detected is different. As far as I can see, the serum dilution at which antiviral antibodies is detectable is similar in WT and FC. Instead, the authors report that the OD is different between the groups at each particular serum dilution. This might indicate a difference in antibody concentration at each dilution, which could be determined using a standard curve or, as mentioned previously, it might indicate a difference in antibody affinity but proof for this conclusion would require additional investigation?

The titres should be shown.

> We revised our text to accommodate the reviewer's concern. Importantly, our overall conclusion remains unchanged. The new text reads: "However, we found that the O.D. values for IAV-specific IgG2b antibodies were significantly reduced in ICOS FC mice, suggesting lower antibody titers and/or affinity." (Page 7).

Incidentally, a similar misunderstanding of serum titre occurs in the reporting of the ANAs in fig. 3B. ANAs are conventionally reported according to the highest serum dilution at which they remain positive. The MFI of positive cells does not correlate with serum dilution.

> As we mentioned in the previous response, we compared multiple serum dilution conditions (undiluted, 1:5, and 1:10) at the beginning of this project. We found that undiluted samples gave oversaturated nuclear signals, which reduced upon dilution. Thus, we believe that the concentration of ANA is reflected by the strength of fluorescence signal and that the signal strength correlates with dilution in a dynamic range. In our mouse model, 1:10 dilution gave the best contrast between genotypes. Our method gives more quantitative results compared to presentation of percentage of positive cases (with a subjective threshold for positive signal) or relative signal strengths using multiple plus signs (e.g., + vs ++ vs +++). With these, we would like to leave the data presentation as is and let the scientific community judge. We elaborated our ANA analysis in *Method* for clarity (page 24): “We recorded the mean fluorescence intensity of 10 nuclei per slide using the “measure” function in ImageJ. Data are presented as mean +/- SEM for each condition.”

3. Contamination of Tfr with Tfh.

My concern was not that the results would be confounded by Tfh cells in the add-back population, rather that cells called nascent Tfr might be Tfh cells formed in the experimental group after immunisation. The authors have attempted to distinguish between Tfh and Tfr using the reference (add-back) population but without barcoding it is impossible to distinguish rare add-back Tfh and use them to determine in which subset rare true Tfh cells are located. The question is, how can we be sure that cluster 3 (Fig. 4C) are not nascent Tfh cells? This is not resolved by the comparison shown in the right panel of Fig. 4C as this is a comparison between cluster 3 and conventional T cells. The comparison should be between cluster 3 and bona fide Tfh cells.

> We argue that cluster 3 (Fig. 4C) should not be heavily occupied by nascent Tfh cells. We know that Tfh cells in the add-back population is ~2 % (Fig. S5B) and the add-back population is only 10 % of the total cells we analyzed. Thus, the number of Tfh cell are ~10. However, we have 187 cells in cluster 3 in the control sample. Furthermore, there is no other source of Tfh cells since the rest of the sample was stringently sorted to be Foxp3⁺ right before sequencing. Therefore, we are confident that cluster 3 represents Tfr cells as opposed to Tfh cells. We added the following sentences in *Discussion* to acknowledge the potential risk arising from our add-back approach (Page 15): “Of note, we added back Foxp3⁻ conventional CD4⁺ cells as reference into our Treg cells (1:10 ratio). A small number of Tfh cells (~2%) within the add-back population may have similar changes in gene expression profile associated with KLF2 downregulation during Tcon-Tfh conversion. However, our analysis should not have been highly affected by the Tfh cells since Tfh-lineage cells are extremely rare (~0.2 % of the total cells) in our dataset.”

Reviewer #3 (Comments to the Authors (Required)):

The authors addressed all my questions. I recommend accepting their work for publication.

January 20, 2023

RE: Life Science Alliance Manuscript #LSA-2022-01615-TRR

Dr. Woong-Kyung SUH
Montreal Clinical Research Institute
110 avenue des Pins Ouest
Montreal, Quebec H2W 1R7
Canada

Dear Dr. SUH,

Thank you for submitting your revised manuscript entitled "ICOS costimulation is indispensable for the differentiation of T follicular regulatory cells". We would be happy to publish your paper in Life Science Alliance pending final revisions necessary to meet our formatting guidelines.

- please upload your supplementary figures as single files and add the supplementary figure legends to the main manuscript
- please add the Twitter handle of your host institute/organization as well as your own or/and one of the authors in our system
- please add a conflict of interest statement to the main manuscript text
- please consult our manuscript preparation guidelines <https://www.life-science-alliance.org/manuscript-prep> and make sure your manuscript sections are in the correct order
- please use the [10 author names, et al.] format in your references (i.e. limit the author names to the first 10)
- please add a figure callout for Figure 4E and Figure S5A to your main manuscript text

A. FINAL FILES:

B. MANUSCRIPT ORGANIZATION AND FORMATTING:

Sincerely,

January 25, 2023

RE: Life Science Alliance Manuscript #LSA-2022-01615-TRRR

Dr. Woong-Kyung SUH
Montreal Clinical Research Institute
110 avenue des Pins Ouest
Montreal, Quebec H2W 1R7
Canada

Dear Dr. SUH,

Thank you for submitting your Research Article entitled "ICOS costimulation is indispensable for the differentiation of T follicular regulatory cells". It is a pleasure to let you know that your manuscript is now accepted for publication in Life Science Alliance. Congratulations on this interesting work.

DISTRIBUTION OF MATERIALS:

Again, congratulations on a very nice paper. I hope you found the review process to be constructive and are pleased with how the manuscript was handled editorially. We look forward to future exciting submissions from your lab.

Sincerely,
